# RELATIVE DRAWING IDENTIFICATION COMPLEXITY IS INVARIANT TO MODALITY IN VISION-LANGUAGE MODELS

## ABSTRACT

Large language models have become multimodal, and many of them are said to integrate their modalities using common representations. If this were true, a drawing of car as an image, for instance, should map to the similar area in the latent space as a textual description of the strokes that conform the drawing. To explore this in a black-box access regime to these models, we propose the use of machine teaching, a theory that studies the minimal set of examples a teacher needs to choose so that the learner captures the concept. In particular, we apply this to GPT-4V, a multimodal version of GPT-4 that includes support for image analysis, to evaluate the complexity of teaching a subset of objects in the *Quick, Draw!* dataset using two presentations: raw images as bitmaps and trace coordinates in TikZ format. The results indicate that image-based representations generally require fewer segments and achieve higher accuracy when compared to coordinate-based representations. But, surprisingly, for concepts recognized by both modalities, the teaching size ranks concepts similarly across both modalities, even when controlling for (a human proxy of) concept priors. This could also suggest that the simplicity of concepts is an inherent property that transcends modality representations.

## 1 INTRODUCTION

As children, when we transform images of the world into drawings and other simplified sketches, we have the intuition that some objects are simpler than others (Chen & Cook, 1984; Long et al., 2018). For instance, six segments are enough to represent a `house` that everybody can recognize, while a bit more are necessary to represent a `cat`. This intuition is epitomized by some guessing games where one person picks a concept from a card deck and has to draw something quick for their team to identify the concept. We can easily describe and recognize some very simple visual concepts, such as letters, with verbalized descriptions. For instance, the letter `T` is a horizontal segment on top of a vertical segment. However, it is challenging for humans to describe complex shapes as verbal descriptions (Sun & Firestone, 2022) or objects, such as a `cat`, using a series of segments.

However, Large Language Models (LLMs) can identify objects from a textual representation of their coordinates (Bubeck et al., 2023). Thus, we need to find out whether this understanding maps to similar capabilities for the multimodal versions of these models. Also, we do not know whether this is independent of the modality. Here, we are asking two research questions:

- Q1 (*Absolute Invariance*): If we randomly sample a concept from a concept class, $c \in C$, would it take the same number of segments to identify it if represented as a bitmap drawing as if represented as a set of coordinates in textual form?
- Q2 (*Relative Invariance*): If we randomly sample two concepts from a concept class, $c_1, c_2 \in C$, such that each of the two concepts is recognized by both modalities, and $c_1$ requires fewer segments than $c_2$ when represented as a bitmap drawing, will this order prevail when expressed as a set of coordinates in textual form?

It is important that we distinguish the second question from the first. For instance, consider $c_1$ is a `house` and $c_2$ is a `cat`. Following the example in Figure 1, if a `house` is easier than a `cat` when using the bitmap of the drawing (top of the figure), is it also easier when represented as

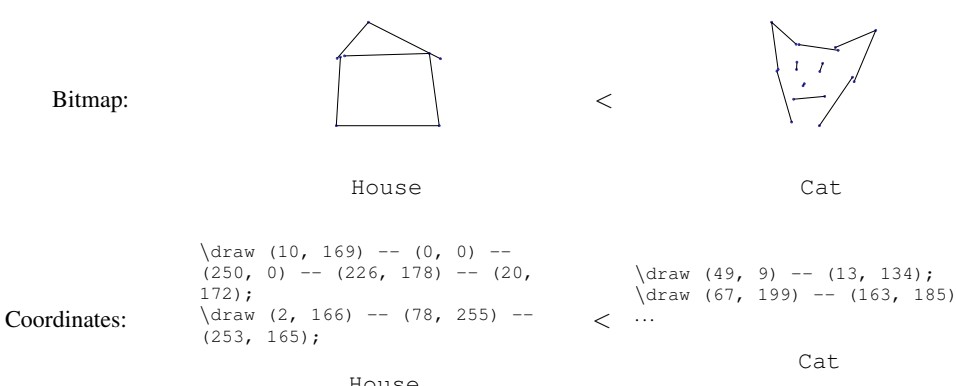

Figure 1: In this paper, we address two research questions. First, Q1 (absolute invariance): When using a vision-language model, are bitmaps (top) equally efficient representations for drawings than coordinates (bottom)? The second question is Q2 (relative invariance): Are the orders (left vs. right) of simplicity preserved across modalities?

segment coordinates (bottom of the figure)? This question Q2 is different from Q1, which refers to whether a concept represented with a bitmap drawing is easier or harder to recognize than the same concept as coordinates in text. Question Q2 is about the ranking, the *relative invariance*. Note that we are not comparing with photographic images of the object since other features would come into play. For instance, a `tiger` is mostly recognized (or distinguished from other felines) by its striped texture rather than by its shape. Such distinctions are particularly evident in machine vision systems (Geirhos et al., 2023). In the rest of this work, when addressing relative invariance, we assume that the two concepts in question have been recognized by both modalities.

However, how will we determine the notion of simplicity of a concept from its drawings? The idea we pursue in this paper is based on the field of machine teaching (Zhu et al., 2018), and in particular, the notion of teaching minimality. A concept is as simple as a teacher can communicate the concept to a learner with as little information as possible. This captures our intuition that a `house` needs six segments while a `cat` needs many more. Given a concept, the teacher thus faces the problem of finding the simplest drawing in terms of the number of straight-line segments—the teaching size—that enables the learner to consistently recognize the concept over a certain number of attempts. We use two different types of language representations (bitmaps of the drawing and coordinates in TikZ code) to present the concepts to the learner. The Generative Pretrained Transformer (GPT)-4V model (Achiam et al., 2023) is employed as the "learner."

It is also important to note that priors play a role in machine teaching. When in doubt, the learner will more likely associate the evidence with the most common concept (e.g., a `house` is more common than an `envelope`). Accordingly, a Bayesian prior will be used to disentangle this effect when looking at the concept simplicity rankings.

The contributions of this paper are:

- A novel machine teaching framework for evaluating the complexity of concepts, which can be applied to drawings in coordinate- and image-based modalities.
- Use of the teaching size specifically to evaluate how simply and effectively the concept can be taught across both modalities.
- A comparison of the effectiveness of both modalities on GPT-4 by focusing on the number of concepts identified, accuracy, frequency of errors, and teaching size.
- A way to disentangle the effect of the learner's prior knowledge in the concept identification task.

These contributions are generic and can be applied to other problems and modalities. In our particular case, we show that bitmaps are more efficient than coordinates, but surprisingly, the order of complexity between the concepts is preserved. This suggests that either the representations of both modalities are tightly connected in the latent space of the model or the simplicity of concepts is an inherent property that transcends modalities.

## 2 RELATED WORK

**Drawing (or Sketches) Recognition**   Eitz et al. (2012) were the first, to our knowledge, to provide a dataset of human drawings. Their dataset includes 250 concepts and 20,000 drawings. In the same work, they introduced a support vector machine model to recognize these drawings and observed that humans outperformed its performance. Since then, artificial intelligent models has been closer or even higher than the accuracy of human classification for drawing recognition (e.g., Schneider & Tuytelaars 2014; Yu et al. 2015; Zhang et al. 2020).

Using the *Quick, Draw!* dataset, Ha & Eck (2017) propose sketch-rnn, a generative model designed to create drawings of common objects that resemble those drawn by humans. A similar version of this model has also shown capabilities in drawing recognition (Bajaj, Payal, 2017). Other neural approaches studied for this task include convolutional neural networks (Kabakus, 2020), and graph neural networks applied over drawings represented as graphs (Xu et al., 2022).

**Drawing (Recognition) Capacities of GPT-4**   Sharma et al. (2024) assess the visual abilities of different language models (including GPT-4). They conduct experiments that prompt the models to create code that draws images based on text descriptions and improve image generation code iteratively through text feedback. Additionally, and of particular relevance to our research, the authors evaluate the model's ability to recognize visual concepts from human drawings converted into code. They arrive at two important conclusions: (a) language models, such as GPT-4, possess limited ability to recognize concepts represented in code, and (b) these models sometimes fail to recognize concepts that they can accurately draw. Note that the authors addressed the problem as a multi-class classification problem. Moreover, the online interface utilized for collecting human drawings is confined to specific components and shapes, such as ellipses. This limitation might restrict the ability of participants to express more complex drawings fully.

In their initial experiments with GPT-4, Bubeck et al. (2023) presents an example of drawing generation, showcasing text-to-image capabilities using TikZ. They show tasks such as GPT-4 drawing a unicorn and constructing TikZ code through a multi-step prompt process. In another study, Pourreza et al. (2023) introduce the *Painter*, a modified LLM that creates drawings using virtual brush strokes based on user-provided text descriptions, with results indicating that Painter can effectively generate, complete, and modify drawings following textual prompts. Additionally, Cai et al. (2023) evaluated GPT-4's ability to understand visual data in SVG format across various visual tasks, including image classification, visual reasoning, and image generation, concluding that GPT-4 possesses the capacity to understand and generate visual content.

**Machine Teaching**   Machine teaching is a research area that focuses on identifying the optimal set of examples that allow a learner (e.g., a human or a machine) to identify a given concept (Zhu et al., 2018). To illustrate the underlying idea of machine teaching, assume the teacher wants the learner to identify the concept of prime numbers. To achieve this, the teacher uses the set $S_1 = \{2, 3, 5, 7, 11, 13\}$ and succeeds. However, would it not be enough for the learner just to see the smaller set $S_2 = \{19, 23\}$? Of course, that depends on the learner. In general, optimal teaching will depend on the model the teacher has about the learner, but we can also consider that the teacher tries many sets in independent experiments to answer that question.

Machine teaching presents an alternative framework to machine learning (where examples are not chosen but sampled from a distribution) to answer the question of whether some concepts are inherently more complex than others. The connections between machine teaching and computational learning theory are strong; see, e.g., the works by Doliwa et al. (2014) or Moran & Yehudayoff (2016), with machine teaching putting the emphasis on the minimal evidence that distinguishes the concept from all the rest. To determine how easy it is to teach a concept, the teaching dimension (Zhu et al., 2018)—the minimum number of examples the learner needs to identify a concept—was traditionally used. Recently, however, Telle et al. (2019) introduced a new metric named teaching size. This metric puts the focus on the sum of the sizes of the examples needed to identify a concept, rather than only the number of examples.

## 3 METHODS

The drawings used in this work come from the *Quick, Draw!* dataset (Jongejan et al., 2016; Ha & Eck, 2017), which includes over 50 million drawings of 345 concepts. Collected by Google Creative Lab via an interactive game, participants had 20 seconds to draw a concept while a neural network attempted real-time recognition. The dataset, which is publicly available and moderated by Google Creative Lab, is the largest collection of doodles in the world, with contributions from more than 15 million participants. Each drawing in the Simplified Drawing files that we use is stored as vectors of distinct pen strokes, i.e., distinct continuous movements of the pen without lifting. Each stroke $s_i$ is represented by a sequence of $(x, y)$ coordinates $\{(x_{i1}, y_{i1}), (x_{i2}, y_{i2}), \ldots, (x_{in}, y_{in})\}$. Note that each pair of consecutive points in a stroke creates a segment. Additionally, for each drawing, a binary flag $r$ indicates whether the game's neural network correctly recognized the concept.

The following sections discuss our selection of concepts and the corresponding drawings. They also introduce the learner, the proposed machine teaching setting, and a set of experiments we carried out before testing this framework, which we call altogether *pre-framework experiment*.

### 3.1 TEACHING SIZE

Let $D$ denote an infinite space of possible drawings (and their simplifications, as will be explained later), and let $C$ be a set of concepts. We use $D_c$ to denote all the drawings of a concept $c \in C$. For any given concept $c \in C$, in some representation, the objective is to identify the simplest drawing $S \in D_c$ such that a learner $L$ successfully learns $c$ with a probability of at least $\rho$ over $N$ independent trials (i.e., recognition consistency). The *teaching size (TS)* of $c$ can be defined as follows:

$$\text{TS}_{\rho,N}(c) = \min_{S \in D_c} |S| \text{ s.t.} \sum_{1}^{N} \mathbb{1}\left[L(S) = c\right] \geq \rho \cdot N. \tag{1}$$

We argue that a good metric for assessing the simplicity of a given drawing $d$ can be based on the number of segments it contains. This is represented by $|S|$ in the above equation.

### 3.2 CONCEPTS

In our work, if the expected concept is `car` and the identified concept is `police car`, the identification is still considered correct because `police car` is a specific type of `car`. This approach is similar to the one followed by Lamb et al. (2020). This means that if a specific sub-concept, or *hyponym*, is identified, it should still be seen as a correct identification as long as it falls under the more general expected concept. For a concept $c$, such as `car`, we consider a set of hyponyms $h(c)$ that corresponds to a set of concepts with a more specific meaning than $c$, e.g., `police car` belongs to $h(car)$. For this study, we want a set of concepts that ensures that in the set of their hyponyms, there is no overlap, i.e., for any two concepts $c_i, c_j$, we have $h(c_i) \cap h(c_j) = \emptyset$. This rules out certain pairs of concepts available in the *Quick, Draw!*, like `van` and `car`, and it enhances the clarity and robustness of the study.

Thus, we select the following subset of 20 concepts from the 345 concepts available in *Quick, Draw!*, with no overlap among their hyponyms: `apple`, `banana`, `car`, `cat`, `computer`, `cup`, `door`, `envelope`, `fish`, `grass`, `hockey puck`, `house`, `key`, `radio`, `string bean`, `sun`, `sword`, `television`, `The Great Wall of China` and `tree`. In Table 3 in the Appendix, we list each concept from the dataset and the accepted hyponyms that are considered correct. This correspondence is established by human inspection and after the execution of the pre-framework (cf. Sect. 3.6) and the machine teaching framework experiments, with the results then analyzed based on these mappings.

### 3.3 DRAWINGS

After choosing the concepts to study, we only include drawings that the game's neural network correctly identified (i.e., $r = 1$) in our research. For every concept, approximately 50 drawings are selected by a proportional random stratified sampling method (Taherdoost, 2016), based on the number of strokes (this number is approximate, as there may be rounding errors when calculating the

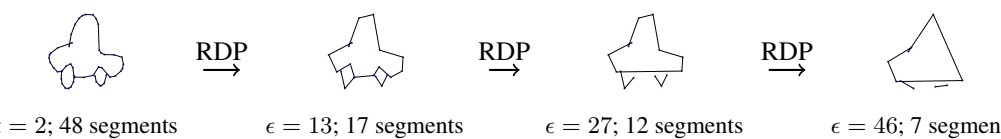

$\epsilon = 2$; 48 segments   $\epsilon = 13$; 17 segments   $\epsilon = 27$; 12 segments   $\epsilon = 46$; 7 segments

Figure 2: Example of a drawing simplification for the concept `car` using the RDP algorithm. As the value of $\epsilon$ increases, the drawings become progressively simpler.

number of samples for each bin according to its proportion.) The bin width was obtained using the minimum bin width between the Sturges's rule and the Freedman Diaconis Estimator. This sampling method ensures that drawings of any concept are represented in a way that reflects the distribution of stroke counts for all correctly identified drawings of that concept in the dataset.

To simplify the drawings in our study, we employ the Ramer–Douglas–Peucker (RDP) algorithm (Ramer, 1972; Douglas & Peucker, 1973) on each stroke $s$ of a given drawing $d$. RDP reduces the number of segments in each stroke while preserving its overall shape. Specifically, given a stroke $s$ with a sequence of points $\{(x_1, y_1), (x_2, y_2), \ldots, (x_n, y_n)\}$, the RDP algorithm iteratively selects the most distant point $(x_d, y_d)$ from the line segment connecting the first and last points of the stroke. If this distance is below a predefined threshold $\epsilon$, then this stroke is simplified to a single segment $\{(x_1, y_1), (x_n, y_n)\}$ on the first and last points. However, if the distance to $(x_d, y_d)$ exceeds $\epsilon$, the algorithm keeps this point and recursively processes the two sequences of points formed by $\{(x_1, y_1), \ldots, (x_d, y_d)\}$ and $\{(x_d, y_d), \ldots, (x_n, y_n)\}$. This ensures that the essential characteristics of the stroke, up to distance $\epsilon$, are preserved. This process continues until all points in the stroke fall within the threshold, resulting in a simplified representation of the stroke with fewer segments. By incrementing the threshold parameter, from an initial value of $\epsilon = 2$ [1], until each stroke is reduced to one segment, we generate simplified versions of each original drawing associated with a given concept $c$, resulting in new drawings $\{d\}_\epsilon \subseteq D_c$. Figure 2 illustrates a drawing simplification.

### 3.4 LEARNER ($L$)

We utilize the GPT-4 model from OpenAI, which is a multimodal LLM capable of processing visual (as per GPT-4V) and language inputs to produce text outputs (Achiam et al., 2023). To conduct the experiments of this work, GPT-4 is accessed using OpenAI's API. Also, we set the temperature parameter $T$ to 1 for the experiments carried out within the machine teaching framework, and we set $T = 0$ for the pre-framework experiment. $T \in [0..2]$ controls the behavior of the model's outputs: the lower $T$ is, the more deterministic (predictable) results it leads to (OpenAI, 2024). Thus, by setting $T = 0$ in the pre-framework experiment, our goal is to obtain deterministic and predictable results, which are essential for creating a consistent baseline of drawings where the concepts were correctly identified. On the other hand, setting $T = 1$ in the experiments of the machine teaching framework is intended to introduce a controlled level of variability, allowing the model to generate diverse outputs while maintaining a degree of predictability.

We consider two different representations for each concept: a visual representation and a text-based representation. Accordingly, we develop and test two prompt templates, one for each modality. For the vision-based modality, the drawings are presented as images generated from the sequence of coordinates (cf. Prompt 1 in the Appendix). For the text-based modality, the pen stroke vectors are coded using the TikZ language (cf. Prompt 2 in the Appendix). Note that both prompts ask for an open-ended answer (not multiple choice), allowing GPT to consider a wide range of possible concepts when identifying a given concept, including those that are not in our 20-concept set.

Let us also briefly discuss the possible issue of contaminated data. Data contamination occurs when language models are tested and evaluated using information from their training data. In this context, this means drawings it has already seen during training (Ravaut et al., 2024). However, in this study, the drawings are consistently simplified using the RDP algorithm. This algorithm alters the coordinate information, thereby modifying the TikZ code and the visual representation.

---

[1]The strokes stored in the Simplified Drawing files of *Quick, Draw!* have already been simplified by the RDP algorithm using $\epsilon = 2$, so this initial value did not simplify any drawing further.

Consequently, we argue that these modified drawings are not part of the training set used to train GPT-4. Therefore, contamination tests are not required for this experiment.

### 3.5 CONCEPT PRIORS

As we argue in the introduction, some concepts, such as a `house`, are more common than others, such as an `envelope`. This sets a strong prior bias, especially in cases of doubt. We obtain these priors for each of the 20 concepts using Google Books Ngram. Google Books Ngram is a tool developed by Google that allows users to analyze how often certain words and phrases appear in an extensive collection of books over time (Google, 2010). Google Books Ngram provides the prior of a given concept as normalized number between 0 and 1, representing the relative frequency of the concept. The rationale for using word frequency from Google Books Ngram as a proxy for human priors lies in the historical and cultural representativity of a corpus. The assumption underlying our approach is that the frequency of specific words and phrases in written text correlates with their prominence in human thoughts, discussions, and collective knowledge at particular times (Tanaka-Ishii & Terada, 2011). Given that GPT models are trained on large text corpora that include books, articles, and other written materials, it is reasonable to assume that the priors derived from Google Books Ngram closely align with the priors embedded in GPT models.

In this study, we use the 2019 English corpus—the latest year accessible when we conducted our experiments—in the Google Books Ngram, extracted with no smoothing factor applied, to serve as the prior. Each concept is analyzed without considering variations in capitalization and is treated strictly as a noun. This approach ensures that, for instance, the concept `fish` is recognized only as the animal instead of the fishing activity, avoiding ambiguity in the prior information.

### 3.6 PRE-FRAMEWORK EXPERIMENT

After selecting the drawings from the concepts for evaluation, we conduct what we call a pre-framework experiment for the generation of a wide range of simplified drawings. Hence, our minimization of Eq. 1 is sufficiently accurate. As already mentioned, the drawings are simplified using the RDP algorithm. The process starts with a threshold of $\epsilon = 2$ on the raw drawings and continues until each stroke in the drawing consists of a single segment. For each $\epsilon$, the learner is prompted using Prompt 1 for visual-based identification and Prompt 2 for text-based identification. Then, based on the completions from the learner, we obtain, by human inspection, the correspondence (between concepts and their respective accepted hyponyms) described in Table 3 in the Appendix, and we analyze the results based on those mappings. The accuracy and frequency of mistakes for each concept are also obtained from the pre-framework experiment.

In total, for the pre-framework experiment, $N = 21,896$ prompts are presented to the learner. We then use the drawings that are correctly identified to test and evaluate the machine teaching framework proposed in Eq. 1, and thus obtain, for each concept, the teaching size.

## 4 RESULTS

### 4.1 CONCEPTS IDENTIFIED

Out of the 20 concepts evaluated, 16 are identified by the learner using images, specifically: `apple`, `banana`, `car`, `cat`, `computer`, `cup`, `door`, `envelope`, `fish`, `house`, `key`, `radio`, `Sun`, `sword`, `television` and `tree`. For coordinates, six concepts are recognized, namely `car`, `cat`, `envelope`, `fish`, `house` and `tree`. In both representations, however, the concepts of `grass`, `hockey puck`, `string bean`, and `The Great Wall of China` are not identified. We hypothesize that not only the complexity but also the prior of each of these latter concepts are behind their failed identification.

The image-based modality is thus more effective than the coordinate-based modality in identifying a broader range of concepts. This observation aligns with the typical human learning patterns, where visual information is often easier to process and understand than abstract text-numerical data.

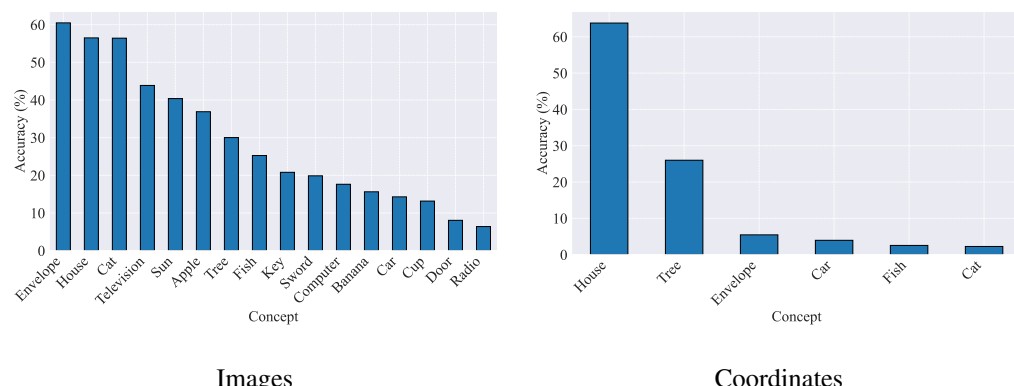

Images                  Coordinates

Figure 3: Accuracy for each concept in the vision-based (images; left) and text-based (coordinates; right) modality representations.

## 4.2 ACCURACY

We begin by evaluating the accuracy on each concept $c$, Accuracy$(c)$, defined here as

$$\text{Accuracy}(c) = \frac{1}{N_c} \sum_{i=1}^{N_c} \mathbb{1}\left[L(S_i) = c\right], \tag{2}$$

where $N_c$ corresponds to the total number of tests (in this case, prompts) conducted on $L$ for the concept $c$ on the pre-framework experiment, with $\{S_i\}_{i=1}^{N_c} \subseteq D_c$.

Figure 3 depicts each concept's accuracy across the two modality representations. We can observe that modalities significantly influence the accuracy levels for the same concepts. For example, the concept envelope achieves an accuracy of $60.48\,\%$ in the image-based modality, while in the coordinate-based modality, it reaches to $5.46\,\%$. This pattern is also observable in car, fish, and cat concepts. Conversely, the accuracy levels between visual and textual modalities are similar for the concepts of house and tree.

For house, it is interesting to note that the concept is identified in over half of all prompts in the coordinate-based modality. One plausible explanation for this is the inherent simplicity and commonality of the house concept. The structure of a house, typically represented by a few straight lines forming a basic geometric shape, can be easily represented using coordinates. This simplicity likely contributes to its higher recognition rate. Additionally, the concept of a house is more common, which may influence the model's priors and contribute to its higher accuracy in both modalities.

We also study the relationship between the number of segments (i.e., complexity) and the accuracy of concept identification for both image- and coordinate-based representations, as shown in Figure 4. For image-based representations, there is a clear positive relationship between the number of segments and accuracy. Starting from an accuracy of around $10\,\%$ in the $(0, 7]$ interval, the accuracy increases steadily, reaching approximately $65\,\%$ in the $(62, 69]$ interval. Conversely, for the text modality, the accuracy remains consistently low across all segment intervals, with values fluctuating between approximately $10\,\%$ and $20\,\%$. This indicates that increasing the number of segments in coordinate-based representations does not significantly improve the accuracy of concept identification (and may even get worse at the end as the description becomes very large).

## 4.3 FREQUENCY OF MISTAKES

Accuracy measures how well the learner has identified the correct concepts. However, the model can also respond with "I don't know" answers (or something that is not a concept), or by identifying a different concept which is incorrect. We focus on the latter case and refer to this performance

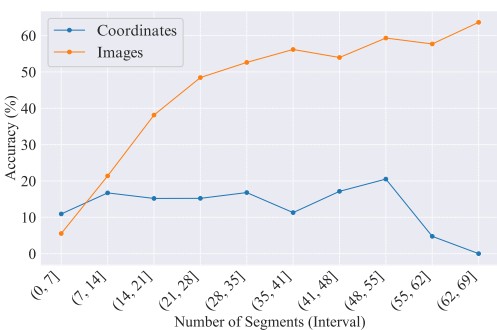

Figure 4: Relationship between the number of segments and accuracy for both modalities.

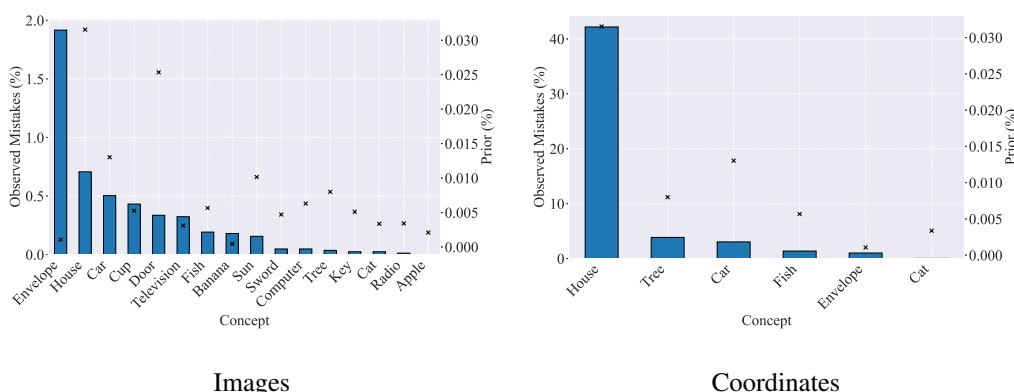

Images          Coordinates

Figure 5: Observed mistakes for each concept in the visual-based modality (images) (left) and text-based modality (coordinates) (right). The little crosses represent the prior probability for each concept.

metric as the *frequency of mistakes* for a given concept $c$, FOM($c$). Formally,

$$\text{FOM}(c) = \frac{1}{N} \sum_{i=1}^{N} \mathbb{1} \left[ S_i \in D_c \wedge L(S_i) \neq c \right], \tag{3}$$

where $N$ is the total number of tests (prompts) conducted on $L$ during the pre-framework experiment. In this study, as already mentioned, $N = 21,896$. We also explore whether there is a relationship between the frequency of mistakes and the prior probability of each concept. We have included in Tables 4 and 5 of the Appendix the confusion matrices for the GPT-4's predictions for both modalities. These tables show how well the model performs across various concepts by detailing the true positives and the frequency of errors for each concept.

Figure 5 shows that the vision modality exhibits a lower percentage of observed mistakes than the text modality. One possible explanation for this difference is that the coordinate-based modality might be selecting answers from a smaller subset of concepts. The marginal row in the confusion matrix for the image-based modality (Table 4) shows that only four concepts were never predicted, even incorrectly, compared to 12 in the text-based modality (Table 5). This hypothesis also aligns with our observation that only six out of the 20 concepts are ever recognized by the text-based modality.

Interestingly, the concept `house` in both modality representations, and `envelope` in only the visual-based modality, show the highest accuracy. However, these also have the highest frequency of mistakes. This indicates that although these concepts are generally easily recognizable, variations in attributes like size and shape may introduce ambiguities that complicate the identification of these concepts. In other words, GPT often guesses these concepts whether they are correct or not. This leads to high accuracy for these concepts but also a high number of observed mistakes.

Table 1: Teaching size, consistency, and priors for the concepts identified by images.

| Concept $c$ | $\text{TS}_{0.5,50}(c)$ | Correct | Incorrect | Prior |
|---|---|---|---|---|
| Envelope | 5 | 50 | 0 | 0.0011 |
| Computer | 6 | 50 | 0 | 0.0063 |
| House | 6 | 50 | 0 | 0.0315 |
| Door | 7 | 45 | 5 | 0.0253 |
| Sun | 7 | 50 | 0 | 0.0101 |
| Sword | 7 | 50 | 0 | 0.0047 |
| Television | 7 | 50 | 0 | 0.0031 |
| Apple | 9 | 50 | 0 | 0.0021 |
| Fish | 9 | 50 | 0 | 0.0057 |
| Banana | 10 | 50 | 0 | 0.0005 |
| Cat | 11 | 50 | 0 | 0.0034 |
| Key | 11 | 50 | 0 | 0.0051 |
| Cup | 13 | 50 | 0 | 0.0052 |
| Tree | 14 | 50 | 0 | 0.0080 |
| Radio | 17 | 50 | 0 | 0.0034 |
| Car | 19 | 50 | 0 | 0.0130 |

Table 2: Teaching size, consistency, and priors for the concepts identified by coordinates.

| Concept $c$ | $\text{TS}_{0.5,50}(c)$ | Correct | Incorrect | Prior |
|---|---|---|---|---|
| Envelope | 5 | 50 | 0 | 0.0011 |
| House | 5 | 50 | 0 | 0.0315 |
| Fish | 15 | 32 | 18 | 0.0057 |
| Tree | 15 | 47 | 3 | 0.0080 |
| Cat | 20 | 50 | 0 | 0.0034 |
| Car | 31 | 50 | 0 | 0.0130 |

When calculating the Pearson correlation between the frequency of mistakes and the prior probability, we obtain a weak correlation of $0.110$ for the images and a strong correlation of $0.949$ for coordinates. This suggests that in textual modality, the learner is more susceptible to responding based on their pre-existing biases when confronted with unfamiliar concepts. In contrast, this tendency is reduced in visual representation.

## 4.4 TEACHING SIZE

To calculate the teaching size for each concept, we set the $T$ to 1, $\rho$ to 0.5, and $N$ to 50, meaning that a correct identification needs to happen at least 25 times out of 50 trials even with some stochasticity in the model. The aim is to determine the simplest drawing for each modality representation that the learner can identify consistently in at least 25 out of 50 trials. We highlight that this procedure is different from the one conducted in the previous sections, where the experiment was part of the pre-framework experiment. We present the results in terms of teaching size for images and coordinates in Tables 1 and 2, respectively. Table 6 of the Appendix shows the original and simplest representations for each concept and modality.

The data suggests that the average teaching size values for coordinates (15.16, SD=8.95) with successful identification (6) are higher than images (10.67, SD=4.78) with successful identification (16). But even if we look at the six concepts that are well identified by coordinates, the means are lower for images. This means there is no absolute invariance, answering our question Q1 in the negative. The number of strokes required to have a concept identified by GPT is higher using textual coordinates than bitmap images.

Furthermore, it is important to highlight a weak, though similar, negative correlation between the teaching size and the prior of each concept across both modalities. The correlation coefficients are $-0.204$ for coordinates and $-0.152$ for images. This may be the case because common concepts

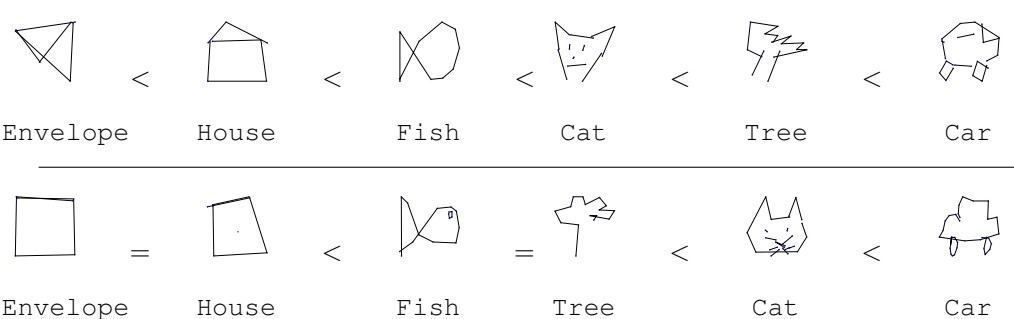

Figure 6: Concepts' teaching size order for images (top) and for coordinates (bottom).

are geometrically simpler, but it is more likely that this is because they are more common and their accuracy is higher; in the same way, we saw that the false positives for these concepts are higher.

Interestingly, however, teaching size ranks concepts in a similar order between images and coordinates. The order is exactly the same except for cat and tree, with a Kendall rank correlation of approximately 0.867. Even when adjusting for concept priors, by regressing the teaching sizes for both modalities on the priors to isolate their residuals and then calculating the Kendall rank correlation on those residuals, we obtain the same correlation value. This means that we have relative invariance, which answers our question Q2 positively. The similar ranking of concept complexity according to teaching size across both modalities indicates that concepts are inherently easier or harder to teach in a relative way, regardless of the data modality and prior.

## 5 DISCUSSION

In this study, we examined how a multimodal model such as GPT-4 identifies the same concepts in two different modalities: either image- or coordinate-based drawings. Our findings show that images are generally more effective than coordinates for identifying concepts. In particular, using images led to the recognition of more concepts than using coordinates, indicating that images are better suited for teaching concepts to a given learner. This is supported by the higher accuracy and lower frequency of mistakes seen with image-based representations. Moreover, we use the number of segments as the teaching size to measure the complexity of a concept. Our analysis indicates that the teaching size is again more beneficial for images than coordinates (answering question Q1 negatively) but consistently ranks concepts in the same way, regardless of the type of drawing used, even when we account for the learner's priors (answering question Q2 positively). This suggests that some concepts are naturally easier or more difficult to teach, no matter how they are represented.

Our analysis has to be seen in the light of some limitations. (a) The study concentrates on a specific set of concepts, which might affect how well the findings apply to other (eventually more complex) concepts. (b) The study employs GPT-4 as the learning model. Although GPT-4 is powerful, results may be different for other models, and of course, also for human learners, something that is out of the scope of this paper. (c) Our use of the RDP algorithm for drawing simplification simplifies each stroke but does not totally remove any single stroke from the drawing. This should not be much of a limitation as we focus on the simplest drawings. (d) A factor that can influence the teaching size of a concept is the curvature of its drawings, i.e., the amount by which it deviates from a straight line. In this work, we have chosen not to focus on this aspect, but this could be of interest for future works.

Our study shows that the simplest drawings usually correspond to those that humans intuitively think of as less complex, and confirms that the simplest drawings are so across modalities. This gives support to the hypothesis that the representation of concepts in both modalities is tightly connected in the latent space. Some other methods, especially white-box approaches having access to weights or gradients, could give a definitive answer to this hypothesis, but in cases such as GPT-4 or humans, a black-box approach as the one presented in this paper is the practical course of action.

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
