# A APPENDIX

## A.1 PROMPTS UTILIZED IN THIS STUDY

In this work, we use two prompt templates to evaluate the effectiveness of concept identification by the GPT-4V model across two different modalities: vision- and text-based representations.

For the visual modality, the drawings are presented as images generated from the sequence of coordinates. The prompt template for this modality involves showing the model the bitmap image of the drawing and asking it to identify the concept depicted in the image.

For the textual modality, the pen stroke vectors are encoded using the TikZlanguage. This format allows the representation of drawings as a series of coordinates and commands that describe the strokes. The prompt template for this modality involves presenting the model with these TikZ-encoded coordinates and asking it to identify the concept represented by the strokes.

Both prompts are designed to elicit open-ended responses from the model, allowing it to consider a wide range of possible concepts, including those not in the predefined 20-concept set. This approach ensures that the model's identification process is not constrained by a limited set of options, thereby providing a more comprehensive evaluation of its capabilities in both modalities.

---

**Prompt 1:** Prompt template for the vision-based modality.

```
 Your task is to identify a concept drawn by hand.  You will be
provided with an image corresponding to a concept drawn by hand.
Your task is to identify, based on the provided picture, the
concept that someone has attempted to draw.  Please reply only with
the name of the concept.
```

**Image URL:** base 64 encoded drawing ($256 \times 256$)

---

**Prompt 2:** Prompt template for the text-based modality.

```
 Your task is to identify a concept drawn by hand.  You will be
provided a TikZpicture format corresponding to a concept, where
each stroke is indicated by the command 'draw' followed by a series
of points in '(x,y)' format.
The points are connected by straight lines, denoted by '--'.  The
strokes collectively represent a concept.  Below is the TikZpicture
code enclosed within triple backticks:
'''{TikZ code}'''.
Your task is to identify, based on the provided TikZpicture, the
concept that someone has attempted to draw.  Please reply only with
the name of the concept.
```

---

**Example of image for the concept `cat`** The vision-based modality, on the other hand, involves using images created from the sequence of coordinates from the *Quick, Draw!*dataset. These images are produced by plotting the coordinates with a function that defines the image size as $256 \times 256$ pixels. The image is then stored in PNG format.

The following is an example of an image representing the cat, extracted from the *Quick, Draw!* dataset.

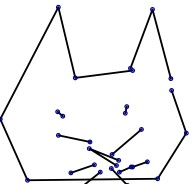

**Example of TikZ code for the concept `cat`** TikZ is a LaTeX package used for creating graphics programmatically. Because of its way of representing drawings through coordinate-based commands, we used TikZ in the text-modality tests.

Each drawing in the *Quick, Draw!* dataset is stored as vectors of distinct pen strokes, represented by sequences of $(x, y)$ coordinates. For each stroke in the drawing, the sequence of points is translated into a `\draw` command. The points are connected using the `--` operator, which denotes a straight line between two points. Each drawing consists of multiple strokes, and each segment is represented by a separate `\draw` command in TikZ.

The following is an example of the TikZ code of the concept `cat`, extracted from the *Quick, Draw!* dataset.

```
1 \draw (181, 30) -- (121, 12) -- (14, 95) -- (0, 161) -- (42, 255) --
2     (73, 213) -- (136, 226) -- (236, 194) -- (242, 230) -- (255, 156) --
3     (218, 38) -- (161, 2) -- (141, 15);
4 \draw (118, 92) -- (76, 118);
5 \draw (119, 81) -- (87, 76);
6 \draw (112, 70) -- (102, 57);
7 \draw (146, 98) -- (192, 107);
8 \draw (151, 76) -- (203, 86);
9 \draw (154, 53) -- (175, 51);
10 \draw (135, 138) -- (137, 71) -- (123, 81);
```

## A.2 ACCEPTED HYPONYMS FOR EACH CONCEPT

Table 3: Accepted hyponyms for each concept. In this study, we establish a set of accepted hyponyms for each concept. A hyponym is a more specific term within a broader category, and for our purposes, identifying a hyponym is considered correct if it falls under the general expected concept. For instance, if the expected concept is `car`, identifying `ambulance` is still correct because it is a specific type of car. This table lists each concept and its accepted hyponyms. These hyponyms are identified in the pre-framework experiment and validated by human inspection.

| Concept ($c$) | Hyponyms ($h(c)$) |
|---|---|
| Apple | Apple with a stem |
| | Apple with a leaf |
| Banana | Banana peel |
| Car | Ambulance |
| | Hovercar |
| | Monster truck |
| | Hovercraft |
| | Floating car |
| | Flying car |
| | Cat whiskers |
| | Cat face |
| | Cat playing with a toy |
| | Origami cat |

Cat

| Concept ($c$) | Hyponyms ($h(c)$) |
| --- | --- |
| | Cat head |
| | Cat and chair |
| | Schrodinger's cat |
| | Cat ear headband |
| | A cat under a table |
| | Geometric fox head |
| | Whiskers |
| | Spider-cat |
| | Sun with cat ears |
| | Sleeping cat |
| | Low poly cat |
| | Cat chasing a laser pointer |
| | Cat playing with a ball of yarn |
| | Cat chasing a mouse |
| Computer | Laptop |
| | Desktop computer |
| | Coffee cup |
| | Coffee mug |
| | Martini glass |
| | Measuring cup |
| | Mug |
| Cup | Takeout coffee cup |
| | Tea cup with tea bag |
| | Teacup |
| | Origami cup |
| | Cup with handle |
| | Paper cup |
| | Car door |
| Door | Door |
| | Open door |
| | Door with a pet door |
| Envelope | Email envelope |
| Fish | Origami fish |
| House | House and tree |
| | Allen key |
| | Key and crystal |
| | Key and lock |
| | Key and screwdriver |
| | Wrench |
| Key | Key and saw |
| | House key |
| | Key and obelisk |
| | Key and tower |
| | Key and keyhole |
| | Arrow key |
| | Hex key |
| Radio | Car radio |
| | Sunburst |
| Sun | Sunlight |
| | Sunshine |
| | Sunrise |
| | Crossed swords |
| | Excalibur |
| Sword | Katana |

| Concept ($c$) | Hyponyms ($h(c)$) |
|---|---|
| | Sword in the stone |
| | Fencing sword |
| | Dagger |
| Television | TV |
| | TV antenna |
| | Television with antenna |
| | Video streaming |
| | Video play |
| | Television playing a video |
| Tree | Christmas tree |
| | Baobab tree |
| | Palm tree |
| | Bonsai tree |
| | Pine tree |
| | Tree with a broken branch |
| | Tree branches |

## A.3 CONFUSION TABLES OF THE CLASSIFICATION

Table 4: Confusion matrix showing the number of times each concept is accurately predicted or misclassified in the visual-based modality (images). Each cell in the matrix represents the count of instances for a specific actual concept versus a predicted concept. The "Other" column shows the number of predictions that do not match any predefined concepts, since GPT-4 is allowed to provide open-ended answers.

| Predicted concept / Concept | banana | fish | string bean | sun | The Great Wall of China | envelope | sword | tree | television | car | hockey puck | grass | cat | house | apple | computer | radio | key | cup | door | Other | Total |
|---|---|---|---|---|---|---|---|---|---|---|---|---|---|---|---|---|---|---|---|---|---|---|
| banana | **78** (15.63%) | 0 (0.00%) | 0 (0.00%) | 0 (0.00%) | 0 (0.00%) | 0 (0.00%) | 0 (0.00%) | 0 (0.00%) | 0 (0.00%) | 0 (0.00%) | 0 (0.00%) | 0 (0.00%) | 0 (0.00%) | 0 (0.00%) | 0 (0.00%) | 0 (0.00%) | 0 (0.00%) | 0 (0.00%) | 1 (0.20%) | 0 (0.00%) | 420 (84.17%) | 499 |
| fish | 0 (0.00%) | **149** (25.25%) | 0 (0.00%) | 0 (0.00%) | 0 (0.00%) | 2 (0.34%) | 0 (0.00%) | 0 (0.00%) | 0 (0.00%) | 0 (0.00%) | 0 (0.00%) | 0 (0.00%) | 0 (0.00%) | 3 (0.51%) | 0 (0.00%) | 0 (0.00%) | 0 (0.00%) | 2 (0.34%) | 0 (0.00%) | 0 (0.00%) | 434 (73.56%) | 590 |
| string bean | 15 (3.55%) | 0 (0.00%) | **0** (0.00%) | 0 (0.00%) | 0 (0.00%) | 1 (0.24%) | 2 (0.47%) | 0 (0.00%) | 0 (0.00%) | 0 (0.00%) | 0 (0.00%) | 0 (0.00%) | 0 (0.00%) | 0 (0.00%) | 0 (0.00%) | 0 (0.00%) | 0 (0.00%) | 0 (0.00%) | 0 (0.00%) | 0 (0.00%) | 404 (95.73%) | 422 |
| sun | 0 (0.00%) | 0 (0.00%) | 0 (0.00%) | **174** (40.37%) | 0 (0.00%) | 0 (0.00%) | 0 (0.00%) | 0 (0.00%) | 0 (0.00%) | 0 (0.00%) | 0 (0.00%) | 0 (0.00%) | 0 (0.00%) | 0 (0.00%) | 0 (0.00%) | 0 (0.00%) | 0 (0.00%) | 0 (0.00%) | 0 (0.00%) | 0 (0.00%) | 257 (59.63%) | 431 |
| The Great Wall of China | 0 (0.00%) | 0 (0.00%) | 0 (0.00%) | 0 (0.00%) | **0** (0.00%) | 1 (0.18%) | 0 (0.00%) | 0 (0.00%) | 0 (0.00%) | 0 (0.00%) | 0 (0.00%) | 0 (0.00%) | 0 (0.00%) | 1 (0.18%) | 0 (0.00%) | 0 (0.00%) | 0 (0.00%) | 0 (0.00%) | 6 (1.07%) | 0 (0.00%) | 553 (98.57%) | 561 |
| envelope | 0 (0.00%) | 0 (0.00%) | 0 (0.00%) | 0 (0.00%) | 0 (0.00%) | **277** (60.48%) | 0 (0.00%) | 0 (0.00%) | 0 (0.00%) | 0 (0.00%) | 0 (0.00%) | 0 (0.00%) | 0 (0.00%) | 0 (0.00%) | 0 (0.00%) | 0 (0.00%) | 0 (0.00%) | 0 (0.00%) | 0 (0.00%) | 0 (0.00%) | 181 (39.52%) | 458 |
| sword | 0 (0.00%) | 0 (0.00%) | 0 (0.00%) | 1 (0.21%) | 0 (0.00%) | 0 (0.00%) | **95** (19.87%) | 0 (0.00%) | 0 (0.00%) | 0 (0.00%) | 0 (0.00%) | 0 (0.00%) | 0 (0.00%) | 1 (0.21%) | 0 (0.00%) | 1 (0.21%) | 0 (0.00%) | 0 (0.00%) | 0 (0.00%) | 0 (0.00%) | 380 (79.50%) | 478 |
| tree | 0 (0.00%) | 0 (0.00%) | 0 (0.00%) | 1 (0.16%) | 0 (0.00%) | 0 (0.00%) | 0 (0.00%) | **187** (30.02%) | 0 (0.00%) | 0 (0.00%) | 0 (0.00%) | 0 (0.00%) | 0 (0.00%) | 3 (0.48%) | 0 (0.00%) | 0 (0.00%) | 0 (0.00%) | 0 (0.00%) | 16 (2.57%) | 1 (0.16%) | 415 (66.61%) | 623 |
| television | 0 (0.00%) | 0 (0.00%) | 0 (0.00%) | 0 (0.00%) | 0 (0.00%) | 5 (0.90%) | 1 (0.18%) | 0 (0.00%) | **243** (43.86%) | 0 (0.00%) | 0 (0.00%) | 0 (0.00%) | 1 (0.18%) | 0 (0.00%) | 0 (0.00%) | 0 (0.00%) | 1 (0.18%) | 0 (0.00%) | 4 (0.72%) | 4 (0.72%) | 295 (53.25%) | 554 |
| car | 0 (0.00%) | 15 (1.91%) | 0 (0.00%) | 0 (0.00%) | 0 (0.00%) | 0 (0.00%) | 0 (0.00%) | 3 (0.38%) | 0 (0.00%) | **112** (14.29%) | 0 (0.00%) | 0 (0.00%) | 0 (0.00%) | 14 (1.79%) | 0 (0.00%) | 0 (0.00%) | 0 (0.00%) | 0 (0.00%) | 0 (0.00%) | 1 (0.13%) | 639 (81.51%) | 784 |
| hockey puck | 0 (0.00%) | 1 (0.19%) | 0 (0.00%) | 0 (0.00%) | 0 (0.00%) | 53 (9.83%) | 1 (0.19%) | 0 (0.00%) | 0 (0.00%) | 0 (0.00%) | **0** (0.00%) | 0 (0.00%) | 0 (0.00%) | 9 (1.67%) | 0 (0.00%) | 1 (0.19%) | 0 (0.00%) | 0 (0.00%) | 0 (0.00%) | 0 (0.00%) | 474 (87.94%) | 539 |
| grass | 0 (0.00%) | 0 (0.00%) | 0 (0.00%) | 1 (0.32%) | 0 (0.00%) | 0 (0.00%) | 0 (0.00%) | 0 (0.00%) | 0 (0.00%) | 0 (0.00%) | 0 (0.00%) | **0** (0.00%) | 0 (0.00%) | 0 (0.00%) | 0 (0.00%) | 0 (0.00%) | 0 (0.00%) | 0 (0.00%) | 0 (0.00%) | 0 (0.00%) | 313 (99.68%) | 314 |
| cat | 0 (0.00%) | 0 (0.00%) | 0 (0.00%) | 10 (1.41%) | 0 (0.00%) | 0 (0.00%) | 0 (0.00%) | 0 (0.00%) | 0 (0.00%) | 0 (0.00%) | 0 (0.00%) | 0 (0.00%) | **400** (56.42%) | 1 (0.14%) | 0 (0.00%) | 0 (0.00%) | 0 (0.00%) | 0 (0.00%) | 3 (0.42%) | 1 (0.14%) | 294 (41.47%) | 709 |
| house | 0 (0.00%) | 0 (0.00%) | 0 (0.00%) | 0 (0.00%) | 0 (0.00%) | 36 (8.20%) | 0 (0.00%) | 0 (0.00%) | 0 (0.00%) | 0 (0.00%) | 0 (0.00%) | 0 (0.00%) | 0 (0.00%) | **248** (56.49%) | 0 (0.00%) | 0 (0.00%) | 0 (0.00%) | 0 (0.00%) | 0 (0.00%) | 0 (0.00%) | 155 (35.31%) | 439 |
| apple | 0 (0.00%) | 0 (0.00%) | 0 (0.00%) | 0 (0.00%) | 0 (0.00%) | 0 (0.00%) | 0 (0.00%) | 0 (0.00%) | 1 (0.16%) | 0 (0.00%) | 0 (0.00%) | 0 (0.00%) | 0 (0.00%) | 2 (0.31%) | **235** (36.89%) | 0 (0.00%) | 0 (0.00%) | 2 (0.31%) | 0 (0.00%) | 0 (0.00%) | 397 (62.32%) | 637 |
| computer | 0 (0.00%) | 0 (0.00%) | 0 (0.00%) | 0 (0.00%) | 0 (0.00%) | 8 (1.40%) | 0 (0.00%) | 0 (0.00%) | 0 (0.00%) | 0 (0.00%) | 0 (0.00%) | 0 (0.00%) | 0 (0.00%) | 3 (0.52%) | 0 (0.00%) | **101** (17.63%) | 0 (0.00%) | 4 (0.70%) | 0 (0.00%) | 7 (1.22%) | 450 (78.53%) | 573 |
| radio | 0 (0.00%) | 0 (0.00%) | 0 (0.00%) | 0 (0.00%) | 0 (0.00%) | 24 (3.74%) | 0 (0.00%) | 0 (0.00%) | 26 (4.06%) | 40 (6.24%) | 0 (0.00%) | 0 (0.00%) | 1 (0.16%) | 20 (3.12%) | 0 (0.00%) | 2 (0.31%) | **41** (6.40%) | 0 (0.00%) | 3 (0.47%) | 0 (0.00%) | 484 (75.51%) | 641 |
| key | 0 (0.00%) | 0 (0.00%) | 0 (0.00%) | 0 (0.00%) | 0 (0.00%) | 3 (0.41%) | 0 (0.00%) | 0 (0.00%) | 0 (0.00%) | 0 (0.00%) | 0 (0.00%) | 0 (0.00%) | 0 (0.00%) | 0 (0.00%) | 0 (0.00%) | 0 (0.00%) | 0 (0.00%) | **151** (20.80%) | 1 (0.14%) | 4 (0.55%) | 567 (78.10%) | 726 |
| cup | 0 (0.00%) | 0 (0.00%) | 0 (0.00%) | 0 (0.00%) | 0 (0.00%) | 19 (3.25%) | 0 (0.00%) | 0 (0.00%) | 0 (0.00%) | 0 (0.00%) | 0 (0.00%) | 0 (0.00%) | 0 (0.00%) | 1 (0.17%) | 0 (0.00%) | 0 (0.00%) | 0 (0.00%) | 0 (0.00%) | **77** (13.16%) | 4 (0.68%) | 484 (82.74%) | 585 |
| door | 0 (0.00%) | 0 (0.00%) | 0 (0.00%) | 0 (0.00%) | 0 (0.00%) | 8 (2.08%) | 0 (0.00%) | 0 (0.00%) | 0 (0.00%) | 2 (0.52%) | 0 (0.00%) | 0 (0.00%) | 0 (0.00%) | 1 (0.26%) | 0 (0.00%) | 0 (0.00%) | 0 (0.00%) | 0 (0.00%) | 2 (0.52%) | **31** (8.05%) | 341 (88.57%) | 385 |
| **Total** | 93 | 165 | 0 | 187 | 0 | 437 | 99 | 190 | 270 | 154 | 0 | 0 | 402 | 307 | 235 | 105 | 42 | 153 | 113 | 59 | 7937 | 10,948 |

Table 5: Confusion matrix showing the number of times each concept is accurately predicted or misclassified in the text-based modality (coordinates). Each cell in the matrix represents the count of instances for a specific actual concept versus a predicted concept. The "Other" column shows the number of predictions that do not match any predefined concepts, since GPT-4 is allowed to provide open-ended answers.

| Predicted concept / Concept | banana | fish | string bean | sun | The Great Wall of China | envelope | sword | tree | television | car | hockey puck | grass | cat | house | apple | computer | radio | key | cup | door | Other | Total |
|---|---|---|---|---|---|---|---|---|---|---|---|---|---|---|---|---|---|---|---|---|---|---|
| banana | 0 (0.00%) | 6 (1.20%) | 0 (0.00%) | 0 (0.00%) | 0 (0.00%) | 0 (0.00%) | 0 (0.00%) | 8 (1.60%) | 0 (0.00%) | 2 (0.40%) | 0 (0.00%) | 0 (0.00%) | 0 (0.00%) | 106 (21.24%) | 0 (0.00%) | 0 (0.00%) | 0 (0.00%) | 0 (0.00%) | 0 (0.00%) | 0 (0.00%) | 377 (75.55%) | 499 |
| fish | 0 (0.00%) | 15 (2.54%) | 0 (0.00%) | 0 (0.00%) | 0 (0.00%) | 1 (0.17%) | 0 (0.00%) | 14 (2.37%) | 0 (0.00%) | 0 (0.00%) | 0 (0.00%) | 0 (0.00%) | 0 (0.00%) | 122 (20.68%) | 1 (0.17%) | 0 (0.00%) | 0 (0.00%) | 0 (0.00%) | 0 (0.00%) | 0 (0.00%) | 437 (74.07%) | 590 |
| string bean | 0 (0.00%) | 0 (0.00%) | 0 (0.00%) | 0 (0.00%) | 0 (0.00%) | 2 (0.47%) | 0 (0.00%) | 1 (0.24%) | 0 (0.00%) | 2 (0.47%) | 0 (0.00%) | 0 (0.00%) | 0 (0.00%) | 115 (27.25%) | 0 (0.00%) | 0 (0.00%) | 0 (0.00%) | 0 (0.00%) | 0 (0.00%) | 0 (0.00%) | 302 (71.56%) | 422 |
| sun | 0 (0.00%) | 8 (1.86%) | 0 (0.00%) | 0 (0.00%) | 0 (0.00%) | 0 (0.00%) | 0 (0.00%) | 28 (6.50%) | 0 (0.00%) | 1 (0.23%) | 0 (0.00%) | 0 (0.00%) | 2 (0.46%) | 138 (32.02%) | 1 (0.23%) | 0 (0.00%) | 0 (0.00%) | 0 (0.00%) | 0 (0.00%) | 0 (0.00%) | 253 (58.70%) | 431 |
| The Great Wall of China | 0 (0.00%) | 0 (0.00%) | 0 (0.00%) | 0 (0.00%) | 0 (0.00%) | 7 (1.25%) | 0 (0.00%) | 18 (3.21%) | 0 (0.00%) | 0 (0.00%) | 0 (0.00%) | 0 (0.00%) | 0 (0.00%) | 85 (15.15%) | 0 (0.00%) | 0 (0.00%) | 0 (0.00%) | 0 (0.00%) | 0 (0.00%) | 0 (0.00%) | 451 (80.39%) | 561 |
| envelope | 0 (0.00%) | 0 (0.00%) | 0 (0.00%) | 0 (0.00%) | 0 (0.00%) | 25 (5.46%) | 0 (0.00%) | 2 (0.44%) | 0 (0.00%) | 0 (0.00%) | 0 (0.00%) | 0 (0.00%) | 0 (0.00%) | 230 (50.22%) | 0 (0.00%) | 0 (0.00%) | 0 (0.00%) | 0 (0.00%) | 0 (0.00%) | 0 (0.00%) | 201 (43.89%) | 458 |
| sword | 0 (0.00%) | 3 (0.63%) | 0 (0.00%) | 0 (0.00%) | 0 (0.00%) | 2 (0.42%) | 0 (0.00%) | 48 (10.04%) | 0 (0.00%) | 3 (0.63%) | 0 (0.00%) | 0 (0.00%) | 0 (0.00%) | 152 (31.80%) | 0 (0.00%) | 0 (0.00%) | 0 (0.00%) | 1 (0.21%) | 0 (0.00%) | 0 (0.00%) | 269 (56.28%) | 478 |
| tree | 0 (0.00%) | 4 (0.64%) | 0 (0.00%) | 0 (0.00%) | 0 (0.00%) | 1 (0.16%) | 0 (0.00%) | 162 (26.00%) | 0 (0.00%) | 6 (0.96%) | 0 (0.00%) | 0 (0.00%) | 0 (0.00%) | 173 (27.77%) | 0 (0.00%) | 0 (0.00%) | 0 (0.00%) | 0 (0.00%) | 0 (0.00%) | 0 (0.00%) | 277 (44.46%) | 623 |
| television | 0 (0.00%) | 3 (0.54%) | 0 (0.00%) | 0 (0.00%) | 0 (0.00%) | 22 (3.97%) | 0 (0.00%) | 2 (0.36%) | 0 (0.00%) | 56 (10.11%) | 0 (0.00%) | 0 (0.00%) | 0 (0.00%) | 386 (69.68%) | 0 (0.00%) | 0 (0.00%) | 0 (0.00%) | 0 (0.00%) | 0 (0.00%) | 0 (0.00%) | 85 (15.34%) | 554 |
| car | 0 (0.00%) | 4 (0.51%) | 0 (0.00%) | 0 (0.00%) | 0 (0.00%) | 1 (0.13%) | 0 (0.00%) | 64 (8.16%) | 0 (0.00%) | 31 (3.95%) | 0 (0.00%) | 0 (0.00%) | 0 (0.00%) | 324 (41.33%) | 0 (0.00%) | 0 (0.00%) | 0 (0.00%) | 0 (0.00%) | 0 (0.00%) | 0 (0.00%) | 360 (45.92%) | 784 |
| hockey puck | 0 (0.00%) | 11 (2.04%) | 0 (0.00%) | 0 (0.00%) | 0 (0.00%) | 15 (2.78%) | 0 (0.00%) | 4 (0.74%) | 0 (0.00%) | 5 (0.93%) | 0 (0.00%) | 0 (0.00%) | 0 (0.00%) | 269 (49.91%) | 0 (0.00%) | 0 (0.00%) | 0 (0.00%) | 0 (0.00%) | 0 (0.00%) | 0 (0.00%) | 235 (43.60%) | 539 |
| grass | 0 (0.00%) | 0 (0.00%) | 0 (0.00%) | 0 (0.00%) | 0 (0.00%) | 0 (0.00%) | 0 (0.00%) | 30 (9.55%) | 0 (0.00%) | 0 (0.00%) | 0 (0.00%) | 0 (0.00%) | 0 (0.00%) | 3 (0.96%) | 0 (0.00%) | 0 (0.00%) | 0 (0.00%) | 0 (0.00%) | 0 (0.00%) | 0 (0.00%) | 281 (89.49%) | 314 |
| cat | 0 (0.00%) | 39 (5.50%) | 0 (0.00%) | 0 (0.00%) | 0 (0.00%) | 0 (0.00%) | 0 (0.00%) | 68 (9.59%) | 0 (0.00%) | 48 (6.77%) | 0 (0.00%) | 0 (0.00%) | 16 (2.26%) | 275 (38.79%) | 1 (0.14%) | 0 (0.00%) | 0 (0.00%) | 0 (0.00%) | 0 (0.00%) | 0 (0.00%) | 262 (36.95%) | 709 |
| house | 0 (0.00%) | 0 (0.00%) | 0 (0.00%) | 0 (0.00%) | 0 (0.00%) | 7 (1.59%) | 0 (0.00%) | 3 (0.68%) | 0 (0.00%) | 4 (0.91%) | 0 (0.00%) | 0 (0.00%) | 0 (0.00%) | 280 (63.78%) | 0 (0.00%) | 0 (0.00%) | 0 (0.00%) | 0 (0.00%) | 0 (0.00%) | 0 (0.00%) | 145 (33.03%) | 439 |
| apple | 0 (0.00%) | 11 (1.73%) | 0 (0.00%) | 0 (0.00%) | 0 (0.00%) | 0 (0.00%) | 0 (0.00%) | 30 (4.71%) | 0 (0.00%) | 5 (0.78%) | 0 (0.00%) | 0 (0.00%) | 0 (0.00%) | 308 (48.35%) | 0 (0.00%) | 0 (0.00%) | 0 (0.00%) | 2 (0.31%) | 0 (0.00%) | 0 (0.00%) | 281 (44.11%) | 637 |
| computer | 0 (0.00%) | 1 (0.17%) | 0 (0.00%) | 0 (0.00%) | 0 (0.00%) | 21 (3.66%) | 0 (0.00%) | 2 (0.35%) | 0 (0.00%) | 49 (8.55%) | 0 (0.00%) | 0 (0.00%) | 0 (0.00%) | 425 (74.17%) | 0 (0.00%) | 0 (0.00%) | 0 (0.00%) | 0 (0.00%) | 0 (0.00%) | 0 (0.00%) | 75 (13.09%) | 573 |
| radio | 0 (0.00%) | 9 (1.40%) | 0 (0.00%) | 0 (0.00%) | 0 (0.00%) | 4 (0.62%) | 0 (0.00%) | 4 (0.62%) | 0 (0.00%) | 107 (16.69%) | 0 (0.00%) | 0 (0.00%) | 0 (0.00%) | 440 (68.64%) | 0 (0.00%) | 0 (0.00%) | 0 (0.00%) | 0 (0.00%) | 0 (0.00%) | 0 (0.00%) | 77 (12.01%) | 641 |
| key | 0 (0.00%) | 1 (0.14%) | 0 (0.00%) | 0 (0.00%) | 0 (0.00%) | 1 (0.14%) | 0 (0.00%) | 58 (7.99%) | 0 (0.00%) | 9 (1.24%) | 0 (0.00%) | 0 (0.00%) | 0 (0.00%) | 309 (42.56%) | 0 (0.00%) | 0 (0.00%) | 0 (0.00%) | 0 (0.00%) | 0 (0.00%) | 0 (0.00%) | 348 (47.93%) | 726 |
| cup | 0 (0.00%) | 42 (7.18%) | 0 (0.00%) | 0 (0.00%) | 0 (0.00%) | 7 (1.20%) | 0 (0.00%) | 10 (1.71%) | 0 (0.00%) | 17 (2.91%) | 0 (0.00%) | 0 (0.00%) | 1 (0.17%) | 313 (53.50%) | 0 (0.00%) | 0 (0.00%) | 0 (0.00%) | 0 (0.00%) | 0 (0.00%) | 0 (0.00%) | 195 (33.33%) | 585 |
| door | 0 (0.00%) | 0 (0.00%) | 0 (0.00%) | 0 (0.00%) | 0 (0.00%) | 14 (3.64%) | 0 (0.00%) | 7 (1.82%) | 0 (0.00%) | 2 (0.52%) | 0 (0.00%) | 0 (0.00%) | 0 (0.00%) | 219 (56.88%) | 0 (0.00%) | 0 (0.00%) | 0 (0.00%) | 0 (0.00%) | 0 (0.00%) | 0 (0.00%) | 143 (37.14%) | 385 |
| Total | 0 | 157 | 0 | 0 | 0 | 130 | 0 | 563 | 0 | 347 | 0 | 0 | 19 | 4672 | 3 | 0 | 0 | 3 | 0 | 0 | 5054 | 10,948 |

### A.4 ORIGINAL AND SIMPLEST DRAWING FOR EACH MODALITY

Table 6: Original and simplest drawing for each concept and modality. For each concept, the table includes both the original drawing and its simplified version, as processed by the Ramer–Douglas–Peucker algorithm. The original drawings are those directly sourced from the *Quick, Draw!* dataset. In contrast, the simplest drawings result from iterative simplification, which reduces the number of segments while preserving the essential characteristics of the concept. This simplified version represents the minimal form that GPT-4, the multimodal large language model used in this study, can still recognize with a high probability (as per the definition of teaching size of this work). By comparing these drawings, we can better understand the inherent simplicity or complexity of each concept and how it translates across visual and textual representations.

| Concept | Original (images) | Simplified (images) | Original (coordinates) | Simplified (coordinates) |
|---------|-------------------|---------------------|------------------------|--------------------------|
| Car | | | | |
| Cat | | | | |
| Envelope | | | | |
| Fish | | | | |
| House | | | | |
| Tree | | | | |

| Concept | Original (images) | Simplified (images) | Original (coordinates) | Simplified (coordinates) |
|---|---|---|---|---|
| Apple |  |  | | |
| Banana |  |  | | |
| Computer |  |  | | |
| Cup |  |  | | |
| Door |  |  | | |
| Key |  |  | | |
| Radio |  |  | | |
| Sun |  |  | | |

| Concept | Original (images) | Simplified (images) | Original (coordinates) | Simplified (coordinates) |
|---|---|---|---|---|
| Sword | | | | |
| Television | | | | |