# OpenReview forum: "Relative Drawing Identification Complexity is Invariant to Modality in Vision-Language Models"
_ICLR.cc/2025/Conference — Submitted to ICLR 2025_

### Official Review · Reviewer_jtY6 · 2024-10-15

**Soundness:** 2
**Presentation:** 3
**Contribution:** 2
**Rating:** 5
**Confidence:** 2

**Summary:**

This work investigates the complexity of teaching visual concepts to large multimodal language models, focusing on how different representations affect the model's ability to learn and identify concepts. The authors assess how efficiently GPT-4V learns objects using machine teaching. The results reveal that image-based representations are generally more effective, requiring fewer segments and yielding higher accuracy. However, the relative ranking of concept complexity remains consistent across both modalities, suggesting that certain concepts are inherently easier or harder to teach, regardless of how they are represented.

**Strengths:**

1. The overall idea of investigating the invariance and complexity of concept representations across modalities is very interesting.
2. The experiment setting is well-presented and easy to understand
3. The description of the experiment setting and research method is very detailed

**Weaknesses:**

1. The experiment is only carried out on GPT-4V, which raises the question of whether the conclusion is specific to the mentioned model or the Vision-Language Models in general. A broader investigation of models such as Gemini [1], LLaVA [2], and CogVLM [3] might strengthen the conclusion of the paper.
2. The experiment uses basic prompts to ask for recognition results without constraining the answering format. The author may consider using constrained decoding or a prompt template with a pre-defined concept set. I believe the concern of the attempts research question can simplify the experiment quite a lot (i.e. no need for finding hyponyms) without sacrificing the recognition capacity.
3. Some of the simplified sketches are not very obvious for human recognition (i.e. the simplest car in Fig 2 does not look like a car; the envelope in fig6 looks like just a square). The experiment only considers basic prompt as a zeroshot inference, which can be challenging for VLM. Something like In-context learning might be helpful to teach the concept to models.

[1] Team, G., Anil, R., Borgeaud, S., Wu, Y., Alayrac, J.B., Yu, J., Soricut, R., Schalkwyk, J., Dai, A.M., Hauth, A. and Millican, K., 2023. Gemini: a family of highly capable multimodal models. arXiv preprint arXiv:2312.11805.
[2] Liu, H., Li, C., Wu, Q. and Lee, Y.J., 2024. Visual instruction tuning. Advances in neural information processing systems, 36.
[3] Wang, W., Lv, Q., Yu, W., Hong, W., Qi, J., Wang, Y., Ji, J., Yang, Z., Zhao, L., Song, X. and Xu, J., 2023. Cogvlm: Visual expert for pretrained language models. arXiv preprint arXiv:2311.03079.

**Questions:**

I'm a bit confused about why GPT4V is called learner. In the experiment, the VLM is presented with sketch and tikz code with fewer segments without an obvious "teaching" process. It looks like a zero-shot inference to me. I would appreciate it if the authors could clarify if I have any misunderstandings about the settings.

In general, I think the idea is very interesting, but the empirical result is a bit weak. In particular, about the number of models tested and the prompt used and inference strategy, such as constrained decoding, pre-defined prompt templates, and in-context learning.

**Details Of Ethics Concerns:**

I don't have any specific concerns about ethics

---

> ### Author Response · Authors · 2024-11-22
>
> **Weaknesses:**
>
> > The experiment is only carried out on GPT-4V...
>
> Thank you for this valuable comment. While we agree that testing additional models like Gemini, LLaVA, and CogVLM could strengthen the generalizability of the conclusions, the primary focus of this paper is the development and validation of the machine teaching framework. GPT-4V was selected as a representative state-of-the-art vision-language model to demonstrate the utility of the framework and explore its application in evaluating cross-modal concept learning.
> We acknowledge that extending the study to include other models is a worthwhile direction for future research. In the revised manuscript, we will clarify the scope of this paper and highlight that the framework is designed to be generalizable and applicable to a broader range of vision-language models.
>
> > The experiment uses basic prompts to ask for recognition...
>
> Thank you for your thoughtful suggestion. While using a prompt template with a pre-defined concept set could simplify the experiment, it would limit the open-ended nature of the evaluation and would complicate the analysis of the class prior. The goal of our study is to assess GPT-4V's ability to identify concepts without biasing or constraining its responses. Allowing open-ended outputs ensures that the evaluation reflects the model’s inherent recognition capabilities and the effect of the model’s prior across both modalities.
>
> > Some of the simplified sketches are not very obvious...
>
> Thank you for your observation. We acknowledge that some of the highly simplified sketches may lack immediate recognizability for humans, and this reflects the challenge of balancing simplicity with semantic fidelity, but it also reflects the relevance of priors. The machine teaching framework aims to identify the minimal representations a learner (in this case, GPT-4) can use to recognize concepts, which may result in sketches that are less interpretable to humans but sufficient for the model.
> In-context learning is indeed a promising approach for teaching concepts to models. However, our study focuses on evaluating the zero-shot recognition capacity of the model, as this setup aligns with the objectives of the machine teaching framework, in measuring the number of segments (teaching size), rather than the number of examples (teaching dimension). We will consider including a discussion in the manuscript about the potential for future work to explore in-context learning or other methods to improve model performance in such cases.
>
> ***
>
> **Questions:**
>
> > I'm a bit confused about why GPT4V is called learner...
>
> Thank you for pointing this out. You are correct that in our experimental setup, GPT-4V operates in a zero-shot inference mode without an explicit "teaching" process. We referred to GPT-4V as a "learner", because this is the traditional terminology within the context of the machine teaching framework even if the “learner” is simply identifying and not actually learning. We will clarify this.
>
> > In general, I think the idea is very interesting, but the...
>
> Thank you for your feedback and for finding the idea interesting. The focus of this work was to introduce and validate the machine teaching framework using a representative vision-language model, GPT-4V, to ensure depth and clarity in our analysis, and to report the surprising finding of the relative preferences, not maximizing identification with different prompt templates or in-context learning. We think the score we have been given focuses on the breadth of the methodology and less so on the novelty of the approach and the scientific findings.
> We agree that testing additional models and exploring alternative inference strategies would provide a more comprehensive evaluation. These are promising directions that we aim to explore in future research. In the revised manuscript, we will explicitly discuss these limitations and suggest how the framework could be extended to address them in subsequent studies.

---

> > ### Comment · Reviewer_jtY6 · 2024-11-24
> >
> > Thank you to the authors for their response. While I agree that the overall idea is interesting from the perspective of understanding VLM capacity, I still feel the empirical findings are not solid enough for me to fully lean toward acceptance. Broader and more solid experiment is needed, including evaluations on additional models, better-curated datasets, and alternative approaches to language model inference. However, as a non-practitioner in machine teaching, I acknowledge my limited knowledge of related concepts and practices, and therefore, I lower my confidence score.

---

### Official Review · Reviewer_WiJ3 · 2024-10-22

**Soundness:** 3
**Presentation:** 2
**Contribution:** 2
**Rating:** 3
**Confidence:** 3

**Summary:**

The paper studies GPT4-V (a multimodal LLM) for the drawing identification task. Specifically, they compare identification accuracy for inputs that vary in terms of complexity (number of drawing strokes) and modality (visual bitmap vs. textual coordinates). For their evaluation, they select a subset of 20 concepts each with 50 associated drawings from the Quick, Draw! dataset. They find that the relative ordering of concept complexity is largely preserved across modalities.

**Strengths:**

- The premise of the paper, or the idea of comparing how vision-language models process analogous image vs. text inputs, is quite interesting and novel.

**Weaknesses:**

- I would like to see some discussion of data filtering / quality checking of the evaluation set, given that RDP is an automated algorithm.
    - Is it possible to conduct an experiment without any RDP simplified images, e.g., for a given concept simply sampling drawings with different numbers of segments from the Quick, Draw! dataset?
- The paper mainly studies the identification accuracy of GPT4-V stratified by different factors (e.g., concept class, modality, level of complexity, relative ranking). It would be nice if it included other lines of inquiry.
    - It would be nice if the study includes additional models or some justification of why GPT4-V is sufficiently representative of “vision-language models” as described in the title. Specifically, it would be nice if the paper included an open-source model, because many aspects of GPT4-V are unknown.
    - It would be nice if the hypothesis in L15 were explored in the paper, i.e., analogous images and textual descriptions “should map to the similar area in the latent space.”

**Questions:**

- Below are a few suggested revisions to improve clarity.
    - In Figure 2, the caption could be revised to explain key terminology, e.g., the “teaching size” (also called “simplicity” / “complexity” of the image) is measured in terms of the number of segments, which is varied by the RDP algorithm.
    - I wish there was an illustration of the input and output to GPT4-V, which showcases the core details of the experiment. For example, the figure could show the input as a bitmap expressed in varying numbers of segments and the output the identification accuracy. Initially, the term “teaching size” made me think there was some learning algorithm at play (e.g., in-context examples).

---

> ### Author Response · Authors · 2024-11-22
>
> **Summary:**
>
> > The paper studies GPT4-V (a multimodal LLM) for...
>
> Thank you for summarizing our work. To clarify, while the final evaluation focused on 20 concepts with 50 drawings each, a total of 21,896 images were tested during the pre-framework experiment. This ensured robust selection of teaching examples and provided the calculation of the teaching size of each concept.
>
> ***
>
> > I would like to see some discussion of data...
>
> Thank you for raising this important point. To ensure the quality of the evaluation set, we filtered the drawings based on the recognition results of the neural network from the Quick, Draw! game. Only drawings that were correctly identified by this neural network were included in our study, providing an automated but reliable quality check for the dataset. While it is possible to use human evaluators to assess the quality of drawings, this approach introduces subjectivity and potential biases. By relying on the game’s neural network, we ensured consistency and scalability in filtering the data while minimizing subjective interpretations.
>
> > Is it possible to conduct an experiment...
>
> Thank you for your suggestion. It is indeed possible to conduct an experiment without using RDP by directly sampling drawings from the Quick, Draw! dataset based on their number of segments. However, we chose to use the RDP algorithm to systematically control the simplification process and ensure consistency across concepts and modalities. Our current sampling technique preserves the natural distribution of simple and complex drawings in the dataset. By stratifying samples to match the ratio of drawings with fewer and more segments, we aimed to reflect the inherent variability in the dataset while ensuring robust comparisons. This approach ensures that both simple and complex drawings are proportionately represented. We appreciate your suggestion, and we will consider adding a note about this in the discussion section.
>
> > The paper mainly studies the identific...
>
> Thank you for this suggestion. While our study focuses on identification accuracy across these factors to establish a clear and controlled understanding of the model’s capabilities with a limited set of research questions to avoid question interaction effects, we agree that exploring additional lines of inquiry could enrich the findings. For example, exploring the models’ capacity for few-shot learning within the same framework could provide valuable insights. We will consider these directions for future work.
>
> > It would be nice if the study includ...
>
> Thank you for your comment. We agree that testing additional models, including open-source ones, could enhance the generalizability of our findings. However, we chose GPT-4V for this study because it represents one of the most advanced and widely recognized vision-language models available. Its strong performance in processing and integrating visual and textual information makes it a suitable representative for exploring cross-modal concept understanding. Including open-source models was considered; however, many existing open-source vision-language models may not yet match the multimodal integration capabilities of GPT-4V. Additionally, due to resource constraints and the scope of this work (the machine teaching framework), we focused on a detailed analysis with a single model to ensure clarity and depth.
> We believe, however, that our methodology is generalizable and can be applied to other models in future research. We will, nevertheless, clarify our rationale for selecting GPT-4V in the manuscript and discuss the potential for applying our framework to other models, including open-source alternatives.
>
> > It would be nice if the hypothesis in L15...
>
> Thank you for pointing this out. However, investigating latent space mappings would require access to the internal representations of the model, which is challenging given the proprietary nature of GPT-4V. One of the main contributions of our work is that machine teaching can be used as a black-box interpretability approach and hence it is applicable to proprietary models, unlike many mechanistic interpretability approaches using white-box models or feature similarity metrics. We appreciate the suggestion and will consider adding a brief discussion of potential future directions in the revised manuscript.
>
> ***
>
> **Questions:**
>
> > In Figure 2, the caption could be revised...
>
> Thank you for the suggestion. We agree that revising the caption would make the figure more accessible and informative. We will follow this suggestion in the revised manuscript.
>
> > I wish there was an illustration of the...
>
> Thank you for the suggestion. We agree that such illustration would enhance clarity and effectively convey the concept of teaching size. We will add such a figure in the revised manuscript to better summarize our work.
>
> ***
>
> Given the clarifications and the planned improvements on the paper, we suggest increasing the score.

---

> > ### Comment · Reviewer_WiJ3 · 2024-11-23
> >
> > Unfortunately, this response to my review makes it challenging to engage in discussion to improve the quality of the paper.
> >
> > I would like to remind the authors that in OpenReview they can "revise their submissions to address concerns that arise," meaning that they are also able to make the promised revisions to the figures during this discussion phase.
> >
> > I have chosen to lower my rating for the following reasons, which have not been addressed in the response above.
> >
> > **1. Limited data filtering / quality checking of the evaluation set.**
> >
> > **It is unclear whether humans can even classify some of the drawings in the evaluation set, which has been automatically generated via RDP.** While the authors note that the data has been automatically filtered by a neural network trained on the Quick, Draw!, I raise this concern due to the examples shown in Table 6 of the Appendix. Specifically, RDP can modify the images so drastically that the simplified images are indistinguishable from other class categories (e.g., the simplified "Computer" and "Door" look almost identical).
> >
> > I also made the concrete suggestion that the authors conduct an experiment without any RDP generated images, to account for this potential limitation, which was not addressed. While I understand that the authors are trying to "reflect the inherent variability in the dataset while ensuring robust comparisons," I think it is still possible to conduct a reasonable evaluation by selecting a smaller subset of images per class, reducing the number of classes, or using a coarser bin width for this specific ablation.
> >
> > **2. Focus is too narrow.**
> >
> > While I understand that the authors have the objective of conducting a "detailed analysis with a single model to ensure clarity and depth," **its focus on identification accuracy of GPT-4V stratified by different factors seems to be narrow to provide broad impact or new insights for the ICLR community.** In particular, the name of the paper suggests that the result applies broadly to "Vision-Language Models," when given the stated research objective it should be revised to "GPT-4V". I do not understand why it would hurt the clarity or depth of the paper to include evaluations on these additional models, as it is common practice to compare against other models as a baseline or study multiple models to observe common trends in a single paper.

---

> > > ### Author Response · Authors · 2024-11-24
> > >
> > > It is our commitment to improve the quality of the paper and revise our paper in due time once all the action points for a new version are clear. Our responses are meant in that direction.
> > >
> > > Regarding the two new points that are behind the new rating:
> > >
> > > 1. [Humans not correctly classifying some of the simplications]. We addressed that issue in one of our responses to reviewer jtY6, which we copy here for this reviewer's convenience: "We acknowledge that some of the highly simplified sketches may lack immediate recognizability for humans, and this reflects the challenge of balancing simplicity with semantic fidelity, but it also reflects the relevance of priors. The machine teaching framework aims to identify the minimal representations a learner (in this case, GPT-4) can use to recognize concepts, which may result in sketches that are less interpretable to humans but sufficient for the model."
> > >
> > > 2. [Experiments with more models]. In our responses we're agreeing that adding more experiments would obviously make the paper more complete, and we're considering doing them. What we're saying is that adapting the setting and conducting experiments for another model is not immediate. Unfortunately this cannot be performed in two weeks given our resources. Then we should prioritize showing new methodologies and findings as soon as we can report them with reasonable support instead of being dragged by a spiral of new models appearing every month. But we have already engaged in quantifying the cost and exploring the funding source to budget for the new experiments, which will have to come after the rebuttal deadline.

---

### Official Review · Reviewer_LwUE · 2024-11-04

**Soundness:** 2
**Presentation:** 2
**Contribution:** 2
**Rating:** 3
**Confidence:** 3

**Summary:**

In this paper, the authors have explored the relative complexity of concept identification across different modalities (bitmap images and coordinate representations) in current large multi-modal models: gpt-4v. The authors introduce machine teaching framework focusing on the minimum information required for identifying concepts in image and coordinate formats. The results suggest that bitmap images are generally more efficient for concept identification than coordinate-based representations.

**Strengths:**

The approach of machine teaching to explore modality-invariant concept complexity is interesting, particularly in comparing GPT-4V's handling of bitmap and coordinate-based representations.

**Weaknesses:**

Most importantly, the practicality of understanding concepts explored in this paper cannot be generalized into the real-world settings, and the findings are not insightful beyond the concept understanding comparison between the image representation and stroke coordinates.

**Questions:**

What are the take-away insights through the analyses and results from the comparison between the bitmap and coordinate-based representations? The current LLMs implicitly learn the concepts for the given explicit supervision: next-word prediction, while the visual perception capabilities are mainly coming through from the pre-trained vision encoder. Can this study partially related with the more real-world scenarios (the way of how the model learns the concepts)?

---

> ### Author Response · Authors · 2024-11-22
>
> **Weaknesses:**
>
> > Most importantly, the practicality of understandi...
>
> Thank you for your comment. While the study focuses on a controlled comparison of concept understanding between image- and coordinate-based representations, we believe its implications extend beyond this specific scenario, and represent an important scientific finding that anticipates potential applications. For example, our findings on teaching size and concept simplicity offer practical value for designing multimodal learning systems that require efficient teaching strategies. These insights can inform applications in education, where adaptive tools could use minimal yet effective representations to teach visual concepts. Additionally, our exploration of modality invariance has potential applications in scenarios where cross-modal consistency is critical, such as accessibility tools that translate between sketches and text or optimizing resource-constrained systems by selecting the most efficient data representations.
> We acknowledge that the broader impact of this work may not be immediately evident from the previous writing of the manuscript. To address this, we will revise the discussion section to emphasize the potential applications of our findings.
> We kindly ask the reviewer to be more explicit in why the findings are not scientifically insightful, according to the answers to the questions below, or adjust the score according to these responses.
>
> ***
>
> **Questions:**
>
> > What are the take-away insights through the analyses...
>
>
> Thank you for this question. The primary takeaways from our analysis are:
>
> - Representation Simplicity Matters: The results show that bitmap representations are more efficient (lower teaching size) and achieve higher accuracy than coordinate-based representations. This suggests that models like GPT-4V are better at leveraging richer, more holistic inputs (e.g., images) than abstract, text-based representations.
>
> - Relative Concept Complexity: Despite the differences in accuracy and teaching size, the rank-order of concept complexity remains consistent across modalities. This highlights that certain concepts are inherently easier or harder for the model to learn, regardless of the representation, suggesting deeper, shared latent structures in how the model learns the concepts or how the integration of modules behaves
>
> - Role of Priors: The influence of priors on concept recognition was evident, particularly in the coordinate-based modality. Concepts with higher priors (e.g., “house”) were more likely to be recognized correctly, even with limited teaching examples. This suggests that models leverage prior knowledge to fill gaps when representation complexity or data quality is lower.
>
> _Relation to Real-World Scenarios:_
>
> This study can inform real-world scenarios by emphasizing how the nature of input representation impacts learning. For example:
>
> - Education and Accessibility: Insights into how models process simplified inputs can help design systems for educational applications, such as tools that teach visual or spatial concepts using minimal representations.
>
> - Resource-Constrained Applications: The findings can guide the development of multimodal systems for real-time or low-resource settings by identifying the most efficient representation for training and inference.
>
> - Concept Transfer Learning: The observed modality invariance in concept rankings suggests that representations capturing fundamental concept properties can potentially improve cross-modal transfer learning in multimodal models.
> We will elaborate on these aspects of the revised manuscript to better link the study’s findings with practical implications.

---

### Official Review · Reviewer_wkWf · 2024-11-05

**Soundness:** 3
**Presentation:** 2
**Contribution:** 2
**Rating:** 5
**Confidence:** 4

**Summary:**

This paper investigates whether MLLMs, specifically GPT-4V, truly understand common representations. The authors leverage machine teaching, a framework focused on identifying the minimum number of teaching cases necessary for a model to learn a concept. They apply this to GPT-4V to assess the complexity of teaching drawing object recognition with two different representations: bitmap images and trace coordinates in TikZ format. The findings show that image-based representations generally require fewer examples and achieve higher accuracy than coordinate-based representations.

**Strengths:**

- The evaluation protocol is technically sound.
- The findings are interesting.

**Weaknesses:**

- The testing model and dataset are limited: in the experiment, only GPT-4V is considered as the model for testing, and the test set is limited to 20 concepts from a specific dataset. Given the variety of multimodal LLMs, including both open-source and proprietary models, the reviewer suggests testing additional models, especially advanced open-source models, to further verify the findings and demonstrate the effectiveness of the proposed protocol.

- Potential applications and impact are unclear: Based on the experimental results shown in this paper, the reviewer is concerned about the potential applications of this work. The experiment tests very simple sketch bitmap images, which is a limited. Also, the discussion of potential applications and impact of this work will be appreciated.

Minor comments:

- The paper's writing should focus more on the motivation. The detailed descriptions of each module are overwhelming for the reader. The reviewer suggests revising the paper in a more concise and precise manner.

**Questions:**

- Line 270 states that the data is not part of the GPT-4V training data. How was this verified?

- When prompting GPT-4, did the authors include the list of concepts in the prompt for GPT-4 to select from?

- Once the teaching size was confirmed, how did you prompt GPT-4 with the instances in the teaching set? Could you please provide more implementation details on how GPT-4 was prompted.

---

> ### Author Response · Authors · 2024-11-22
>
> **Weaknesses:**
>
> > The testing model and dataset are lim...
>
> Thank you for this suggestion. To clarify, while the final evaluation focused on 20 concepts with 50 drawings each, a total of 21,896 drawings were tested during the pre-framework experiment. We understand that testing more models would enhance the generalizability. Our main goal in this study was to introduce and validate the Machine Teaching (MT) framework for assessing concept complexity across modalities, discovering the relative invariance. We wanted to communicate this finding in an expeditious way, and this is what conferences are for, in an area such as AI, where new models appear every month. We agree that including more models would strengthen our results. However, this was not feasible for the submission deadline due to limited resources and the complexities involved in testing both proprietary and open-source systems. The differences in model accessibility, API functionality, prompt interpretation, scoring the outputs and the pre-framework experimental selection would require extensive adjustments to ensure fair comparisons. These adjustments might shift the focus away from the main purpose of our study, which is the MT framework. We believe that future studies will expand on our work. These limitations were described in the Discussion section of our paper.
>
> > Potential applications and impact are...
>
> Thank you for your comment. One potential application is in education. In education, the MT framework could support adaptive teaching tools, identifying the simplest, most effective representations for individual learning needs. For example, it could help design software for teaching geometric concepts or visual reasoning through minimal examples. Additionally, the framework offers insights into how multimodal models like GPT-4V process and generalize across visual and textual inputs, contributing to model interpretability in cross-modal tasks. The evaluation of teaching size and concept complexity could also optimize multimodal training, focusing on efficiency and minimal data requirements. We will discuss these applications and their potential impact in the paper.
>
> ***
>
> **Minor comments:**
>
> > The paper's writing should focus mor...
>
> Thank you for this feedback. We agree that a stronger focus on the motivation and a more concise presentation of the detailed descriptions would improve the paper. We ask the reviewer to adjust the part of the score that may have derived from this misunderstanding. We will revise the manuscript with this in mind.
>
> ***
>
> **Questions:**
>
> > Line 270 states that the data is no...
>
> Thank you for raising this question. The original dataset used in our experiments was significantly altered through the RDP algorithm. This simplification process modifies the coordinate information and visual representations, resulting in transformed versions of the data that are unlikely to have been part of GPT-4V’s training data. We happy to clarify this reasoning in the manuscript.
>
> > When prompting GPT-4, did the author...
>
> Thank you for this question. No, the list of concepts was not included in the prompts provided to GPT-4. Instead, the prompts were designed to allow open-ended responses, enabling the model to identify the concept based solely on the input representation. This approach ensured that the model's recognition relied on its understanding rather than being constrained by or biased towards a predefined list of options. We also highlight here that to evaluate the correctness, we matched the model's responses to a predefined set of acceptable hyponyms for each target concept. We can clarify this further in the manuscript if needed.
>
> > Once the teaching size was confirmed...
>
> Thank you for your question. The process was as follows:
>
> _1. Pre-Framework Experiment:_
>
> Before calculating the Teaching Size (TS), we conducted a pre-framework experiment to evaluate the model's ability to identify concepts across a wide range of simplified inputs. The original drawings were progressively simplified using the RDP algorithm. For each level of simplification, we prompted GPT-4V with the concepts’ representation (bitmap for image-based or TikZ code for coordinate-based), using the following structure:
> - Image: The bitmap image was directly uploaded.
> - Coordinate: The TikZ representation was passed as plain text in the prompt.
>
> _2. Teaching Size Calculation:_
>
> Once the pre-framework experiment determined the minimal representation for consistent recognition, we calculated the TS as the number of segments in the simplest drawing that allowed the model to recognize the concept with at least 50% accuracy over 50 trials. The teaching set was then composed of these minimal representations for each concept. The draws were prompted to GPT following a similar structure as in the pre-framework experiment (both in terms of representations and prompts). We will improve these methodological details in the revised manuscript for clarity.

---

### Author Response · Authors · 2024-11-22
**General comments**

The reviewers are positive about the novelty of the approach and the scientific findings, but we have the impression that the scores are dominated by a request to test more models (more is of course always better). We would like to ask the reviewers to calibrate the actual contribution of the paper in the scores: the use of machine teaching for understanding representation is novel, and can give us important insights about the representations –without opening the box–, with applicability on proprietary models (unlike mechanistic interpretability approaches). The findings, especially the relative preferences, can also lead to significant follow-up papers.

---

### Meta-Review · Area_Chair_5o9J · 2024-12-22

**Metareview:**

The paper tests GPT-4V’s ability to recognize drawings in bitmap versus coordinate form. While bitmaps generally need fewer examples, the relative complexity across concepts is similar in both formats. The study’s machine teaching approach is interesting, but the work relies on a narrow setup (only GPT-4V) and questionable data filtering. No comparisons to other models limit generalizability. Overall, these issues lead to a reject recommendation from the AC.

**Additional Comments On Reviewer Discussion:**

Reviewers questioned the limited model coverage, data-filtering rigor, and real-world applicability. The authors addressed some points but could not provide expanded experiments or broader datasets within the rebuttal. After weighing the clarifications against remaining concerns, the AC made a reject decision.

---

### Decision · Program_Chairs · 2025-01-22

Reject